# Clinicopathologic vs. Molecular Integrated Prognostication of Endometrial Carcinoma by European Guidelines

**DOI:** 10.3390/cancers14030651

**Published:** 2022-01-27

**Authors:** Mikko Loukovaara, Annukka Pasanen, Ralf Bützow

**Affiliations:** 1Department of Obstetrics and Gynecology, Helsinki University Hospital, University of Helsinki, 00290 Helsinki, Finland; ralf.butzow@hus.fi; 2Research Program in Applied Tumor Genomics, Department of Pathology, Faculty of Medicine, Helsinki University Hospital, University of Helsinki, 00290 Helsinki, Finland; annukka.pasanen@hus.fi

**Keywords:** endometrial carcinoma, European Society for Radiotherapy and Oncology, European Society of Gynaecological Oncology, European Society of Pathology, mismatch repair, polymerase-ϵ, p53, The Cancer Genome Atlas

## Abstract

**Simple Summary:**

The European Society of Gynaecological Oncology (ESGO), European Society for Radiotherapy and Oncology (ESTRO), and European Society of Pathology (ESP) published joint guidelines in January 2021 that provide recommendations on all relevant issues of diagnosis and treatment in endometrial carcinoma. Assessment of prognosis and adjuvant therapy decisions are based on classification of endometrial carcinomas into five risk groups with specific clinicopathologic features. Integration of molecular classification, originally described by The Cancer Genome Atlas, is encouraged for a more personalized risk assessment when molecular tools are available. We found that clinicopathologic and molecular integrated risk groups were similarly associated with distinct prognoses. The p53 abnormal molecular subgroup and mismatch repair deficient molecular subgroup were associated with poor survival within clinicopathologic low-risk and high-intermediate-risk carcinomas, respectively. Molecular classification caused a risk-group shift in 6–7% of patients. Comprehensive molecular classification was needed in 40% of patients for molecularly directed adjuvant therapy.

**Abstract:**

This was a retrospective study of 604 patients with endometrial carcinoma, classified into ESGO-ESTRO-ESP 2021 clinicopathologic and molecular integrated risk groups. The Proactive Molecular Risk Classifier for Endometrial Cancer (ProMisE) and Leiden classifier were employed for molecular classification. Median follow-up time was 81 months. Clinicopathologic and molecular integrated risk groups were similarly associated with distinct prognoses (*p* < 0.001). Disease-specific survival was similar for all molecular subgroups within clinicopathologic intermediate-risk, high-risk, and advanced/metastatic groups. In contrast, the p53 abnormal subgroup (hazard ratio 9.1, 95% confidence interval 2.0–41; *p* = 0.004) and mismatch repair deficient subgroup (hazard ratio 3.5, 95% confidence interval 1.2–10; *p* = 0.024) were associated with disease-related death within clinicopathologic low-risk and high-intermediate-risk carcinomas, respectively. A risk-group shift occurred in 6.0% (36/604) and 7.4% (38/515) of patients classified by ProMisE and Leiden, respectively (*p* = 0.341). Of the 36 patients shifted in the ProMisE cohort, 27 were upshifted and 9 downshifted. Based on the Leiden classifier, polymerase-ϵ sequencing could be omitted in 60% (311/515) of patients without affecting the risk-group assessment. ESGO-ESTRO-ESP 2021 guidelines provide a platform for risk classification in future trials on molecularly directed treatment of endometrial carcinoma.

## 1. Introduction

Updated guidelines for endometrial carcinoma by the European Society of Gynaecological Oncology (ESGO), European Society for Radiotherapy and Oncology (ESTRO), and European Society of Pathology (ESP) were published in January 2021 [1]. As an essential step toward personalized therapy, the updated guidelines encourage molecular classification in all endometrial carcinomas, especially high-grade tumors. When molecular classification tools are not available, postsurgery risk stratification should be based on traditional clinicopathologic features.

Outcome analyses have confirmed that there is a molecular subgroup of endometrial carcinomas with an excellent prognosis—i.e., the polymerase-ϵ (*POLE*) ultramutated tumors—and a group with a poor prognosis—i.e., the copy-number high tumors [2]. Microsatellite-instability hypermutated and copy-number low tumors have an intermediate prognosis [2]. Based on this knowledge, ESGO-ESTRO-ESP 2021 guidelines propose treatment intensification in early-stage copy-number high carcinomas, and treatment de-escalation in early-stage *POLE* ultramutated carcinomas, regardless of traditional clinicopathologic risk factors [1].

Here, we retrospectively implemented the ESGO-ESTRO-ESP 2021 clinicopathologic and molecular integrated risk groups in an unselected cohort of 604 women with endometrial carcinoma. Our purpose was to compare risk group outcomes with and without molecular knowledge, and to assess the frequency of shift between risk groups with integration of molecular classification. Moreover, because the prognostic effect of molecular subgroups may be altered by traditional risk factors [2,3], we compared outcomes for the molecular subgroups separately within each clinicopathologic risk group.

## 2. Materials and Methods

This was a retrospective study of patients who underwent surgical treatment for stage I–IV endometrial carcinoma at the Department of Obstetrics and Gynecology, Helsinki University Hospital, between 1 January 2007 and 31 December 2012. Clinicopathologic data were abstracted from institutional medical and pathology records. Lymphovascular space invasion (LVSI) was defined according to a three-grade system as follows: none (no LVSI), focal (presence of a single focus around the tumor), and substantial (multifocal or diffuse arrangement of LVSI or the presence of tumor cells in five or more lymphovascular spaces) [1]. Stage was determined according to the International Federation of Gynecology and Obstetrics guidelines revised in 2009 [4]. The final cohort consisted of patients with a successful molecular characterization of their primary tumors.

Patients with early-stage endometrioid carcinoma with high-risk features generally received either vaginal brachytherapy or whole pelvic radiotherapy as adjuvant treatment. Vaginal brachytherapy was preferred in those who underwent surgical nodal assessment. Patients with nonendometrioid or advanced-stage endometrioid carcinoma usually received combination treatment with chemotherapy and radiation. Chemotherapy and whole pelvic radiotherapy were typically delivered sequentially. Paclitaxel/carboplatin doublet was the standard chemotherapy regimen.

Disease-specific survival was calculated as the time from surgery to death from endometrial carcinoma. Cause of death was mainly based on medical records. Missing data were complemented from death certificates provided by Statistics Finland.

We constructed a tissue microarray on primary tumor samples as previously described [5], and performed immunohistochemistry for p53 and mismatch repair (MMR) proteins MLH1, MSH2, MSH6, and PMS2. Abnormal p53 staining was defined as strong and diffuse nuclear staining or entirely negative (“null”) staining in carcinoma cells. MMR protein status was considered deficient when a complete loss of nuclear expression in carcinoma cells of one or more MMR proteins was observed. *POLE* exonuclease domain mutation screening of hot spots in exons 9, 13, and 14 was performed by direct sequencing [6]. Only samples with high-quality sequence for all the four *POLE* hot spots examined were included in the study.

We employed two classifiers; i.e., the Proactive Molecular Risk Classifier for Endometrial Cancer (ProMisE) [7] and Leiden [8], to recapitulate the molecular subgroups of endometrial carcinoma originally described by The Cancer Genome Atlas (TCGA) research network [9]. Tumors were classified as “no specific molecular profile” (NSMP, surrogate to copy-number low in the TCGA classification system [9]); mismatch repair deficient (MMRd, surrogate to microsatellite unstable hypermutated); p53 abnormal (p53abn, surrogate to copy-number high); and *POLE* mutant (*POLE*mut). In the ProMisE classifier [7], tumors are classified in a stepwise fashion. The first subgroup assignment is based on MMR status. Tumors with intact MMR proteins undergo *POLE* mutational analysis and *POLE* wild-type (wt) tumors are classified as p53abn or p53wt/NSMP.

In the Leiden classifier, all molecular markers are determined for each sample, and cases with multiple molecular alterations are excluded from the classification [8]. In contrast to the original Leiden protocol, we included cases with multiple classifying alterations. *POLE*mut–MMRd and *POLE*mut–p53abn tumors were classified as *POLE*mut [10], and MMRd–p53abn tumors as MMRd [11].

The patients were stratified into both ESGO-ESTRO-ESP 2021 clinicopathologic and molecular integrated risk groups [1]. Categorical variables were compared by Pearson’s χ^2^ or two-sided Fisher’s exact test. Survivals were determined using univariable Cox regression analyses and the Kaplan–Meier method. Differences between groups were compared using the log-rank test. Statistical significance was set at *p* < 0.05. Data were analyzed using the Statistical Package for the Social Sciences version 25 software (IBM Corp., Armonk, New York, NY, USA).

## 3. Results

A total of 604 women were stratified into ESGO-ESTRO-ESP 2021 risk groups with clinicopathologic factors alone, and with integration of the ProMisE molecular classifier. Median follow-up time was 81 months (range 1–136). The basic characteristics of the study population are summarized in Table 1. The distribution of ESGO-ESTRO-ESP 2021 risk groups among the study population is shown in Figure 1. The low-risk group was most common, comprising >40% of all cases, whereas the advanced/metastatic group was least common (4.0%).

Kaplan–Meier disease-specific survival analyses confirmed stratification of endometrial carcinomas into distinct prognostic groups by ESGO-ESTRO-ESP 2021 clinicopathologic and molecular integrated systems (Figure 2). Pairwise comparisons showed more overlap in molecular integrated risk groups (Figure 2).

Table 2 shows univariable Cox regression disease-specific survival analyses for ESGO-ESTRO-ESP 2021 clinicopathologic risk groups with the ProMisE classifier as the dependent variable. There was only one *POLE*mut case (stage IA serous carcinoma) in the clinicopathologic high-risk group, and none in the advanced/metastatic group. Outcomes were similar for all molecular subgroups within clinicopathologic intermediate-risk, high-risk, and advanced/metastatic groups. In contrast, p53abn and MMRd were associated with poor outcome within clinicopathologic low-risk and high-intermediate-risk groups, respectively.

Molecular characterization by the Leiden classifier was successful on 515 tumors. Twenty cases (3.9%) displayed multiple molecular alterations. Four cases were classified as *POLE*mut tumors [10]: three displayed *POLE*mut and either MMRd or p53abn, and one had all three molecular alterations. Sixteen cases were classified as MMRd tumors [12], displaying both MMRd and p53abn.

Table 3 shows the shift between prognostic risk groups with implementation of molecular classification. Thirty-six (6.0%) and 38 (7.4%) patients were shifted between risk groups by the ProMisE and Leiden molecular integrated schemas, respectively (*p* = 0.341). Of the 36 patients shifted in the ProMisE cohort, 27 were upshifted and 9 downshifted. The occurrence of shift was similar for the two classifiers in all risk groups. Shifts mostly occurred in the high-intermediate risk group.

Based on the Leiden classifier, we assessed the proportion of patients in whom *POLE* sequencing could be omitted without causing a risk-group shift; i.e., those with either clinicopathologic low-risk carcinoma and normal p53 staining (208/515), or stage III–IV carcinoma (103/515). Altogether, *POLE* sequencing could be omitted in 60% (311/515) of patients.

## 4. Discussion

In this study, we meticulously replicated the ESGO-ESTRO-ESP 2021 clinicopathologic and molecular integrated risk groups of endometrial carcinoma in a large, unselected cohort. We confirmed distinct outcomes for the five risk groups with both approaches.

We also compared outcomes for the molecular subgroups separately within each clinicopathologic risk group. As expected, p53abn was associated with poor disease-specific survival within clinicopathologic low-risk carcinomas. In contrast, MMRd was associated with poor survival within clinicopathologic high-intermediate-risk carcinomas. This may be associated with the notion that the prognosis of the MMRd subgroup overlaps with NSMP, but is worsened by unfavorable clinicopathologic factors [2] that are enriched in the high-intermediate risk group [1]. The finding could also be explained by a poor response of MMRd carcinomas to adjuvant radiotherapy [12]. Although a contrasting finding has been reported [13], this study may not be similarly applicable in the context of TCGA because tumors were dichotomously categorized into MMRd and MMR proficient subgroups, with the latter including NSMP, *POLE*mut, and p53abn cases.

Molecular subgroups were associated with rather modest hazard ratios for disease-related death within the different clinicopathologic risk groups, which emphasizes the need to develop molecular subgroup-specific prognostic tools in endometrial carcinoma. The L1 cell adhesion molecule (L1CAM) may be one such example, as it has been shown to be an independent predictor of worse disease-specific survival within the NSMP subgroup [14]. L1CAM is included as a risk variable in the ongoing PORTEC-4a trial to evaluate vaginal recurrence after adjuvant treatment or observation based on molecular-integrated risk profile in women with early-stage endometrial carcinoma [15].

By using the ProMisE schema for molecular classification, 6.0% of patients were downshifted or upshifted to another risk group due to a pathogenic *POLE* mutation or abnormal p53 staining, respectively. In an earlier study, risk groups were discordant in 6.6% (39/594) of patients classified with the 2016 clinicopathologic and 2021 molecular integrated systems [16]. In the 2016 system, LVSI is not graded, and its presence is less weighted in risk assessment, whereas cervical stromal invasion is weighted more, and the absence of myoinvasion is not considered in risk assessment of nonendometrioid carcinomas [17].

Molecular classification mainly caused an upshift in risk grouping, usually to high-risk of p53abn carcinomas with myoinvasion (26/604 based on ProMisE), and occasionally to intermediate-risk of p53abn cases without myoinvasion (1/604). As per the 2021 guidelines, adjuvant chemotherapy with or without whole pelvic radiotherapy is recommended in high-risk patients, whereas vaginal brachytherapy is the primary adjuvant therapy of choice in intermediate-risk patients [1].

Downshift was less common, but was observed in 9/604 patients with *POLE* mutation; adjuvant therapy could be omitted in all of them [1]. Proportions of upshifted and downshifted patients were similar for the ProMisE decision tree analysis and the Leiden comprehensive molecular classifier. Thus, both methods appear feasible for molecular classification. It should be noted, however, that ProMisE identifies *POLE*mut–MMRd double classifiers not as *POLE*mut [10] but as MMRd, some of which are then categorized as intermediate- or high-intermediate-risk carcinomas with vaginal brachytherapy or whole pelvic radiotherapy as the adjuvant therapy of choice [1]. However, these double classifiers are uncommon (1/515 in our sample). Similarly, MMRd–p53abn cases (16/515) cannot be identified by ProMisE. They are identified as MMRd, which is the prognosis-determining alteration for this double classifier [11]. This omits the need to identify MMRd–p53abn cases in clinical practice.

Even with comprehensive molecular testing as the principal classifying method, it may be possible to reduce *POLE* sequencing without affecting the risk-group assessment. Abnormal p53 staining was a rare finding in low-risk carcinomas (10/218 based on Leiden), but was associated with poor survival. It seems reasonable to restrict *POLE* sequencing to clinicopathologic low-risk carcinomas with abnormal p53 staining, whereby *POLE*mut–p53abn double classifiers can be identified. *POLE* testing can be further reduced by omitting it in advanced (stage III–IV) carcinomas, in which adjuvant therapy decisions are not altered by molecular classification [1]. Altogether, *POLE* sequencing could be omitted in 60% of endometrial carcinomas.

It should be noted that with the lack of randomized trials on molecularly classified endometrial carcinomas, ESGO-ESTRO-ESP 2021 adjuvant therapy guidelines are mainly based on the intrinsic survival differences between molecular subgroups [2,9]. Knowledge of the relationship between molecular subgroups and benefit from adjuvant therapy was enhanced by Léon-Castillo et al., who compared chemoradiotherapy versus whole pelvic radiotherapy for each molecular subgroup using tissue samples from the PORTEC-3 trial [18]. The participants mainly corresponded to ESGO-ESTRO-ESP 2021 high-risk patients. Adjuvant chemotherapy improved recurrence-free survival for p53abn carcinomas. Of them, 73% were nonendometrioid or mixed, and 34% were stage III. Patients with NSMP and MMRd carcinomas did not benefit from adjuvant chemotherapy. Patients with *POLE*mut carcinomas had an excellent recurrence-free survival in both trial arms. The knowledge gap of optimal molecularly directed adjuvant therapy remains most evident for intermediate- and high-intermediate-risk groups that comprise about one-third of all endometrial carcinomas.

## 5. Conclusions

ESGO-ESTRO-ESP 2021 clinicopathologic and molecular integrated risk groups are associated with distinct prognoses. Molecular classification causes a risk-group shift in a meaningful proportion of patients. *POLE* sequencing, the most laborious component of molecular classification, may safely be omitted in 60% of patients. Clinicopathologic risk factors may differently modify the prognostic impact of molecular subgroups. This emphasizes the need for adjuvant therapy trials in which patients are randomized to treatment arms separately within each molecular subgroup.

## Figures and Tables

**Figure 1 cancers-14-00651-f001:**
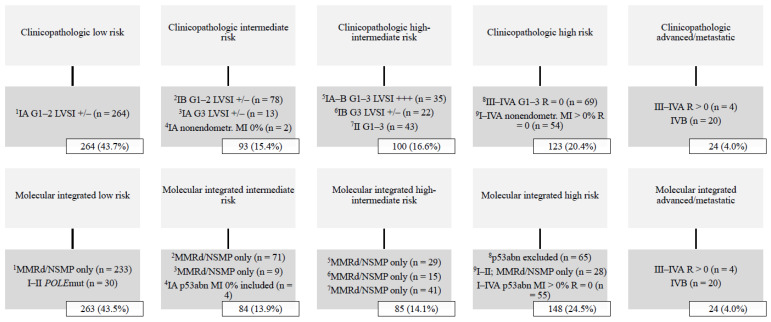
Distribution of ESGO-ESTRO-ESP 2021 risk groups by the ProMisE classifier (n = 604). Abbreviations: G, histologic grade; LVSI +/−, lymphovascular space invasion focal or negative; LVSI +++, lymphovascular space invasion substantial; MI, myometrial invasion; MMRd, mismatch repair deficient; NSMP, no specific molecular profile; *POLE*, polymerase-ϵ; p53abn, p53 abnormal; R, residual tumor.

**Figure 2 cancers-14-00651-f002:**
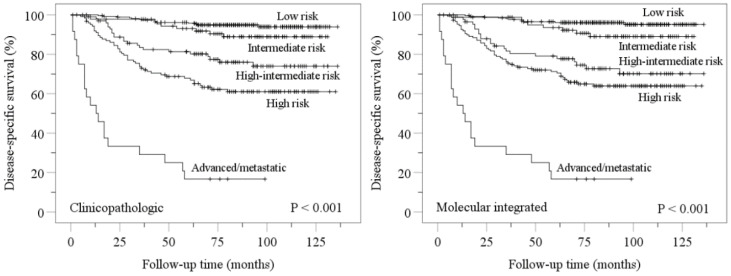
Kaplan–Meier estimations for disease-specific survival according to ESGO-ESTRO-ESP 2021 clinicopathologic and molecular integrated risk classification systems. Nonsignificant *p* values for pairwise comparisons were as follows: clinicopathologic low vs. intermediate, *p* = 0.120; molecular integrated low vs. intermediate, *p* = 0.056; molecular integrated high-intermediate vs. high, *p* = 0.207.

**Table 1 cancers-14-00651-t001:** Clinicopathologic data (n = 604).

Molecular Subgroup	Number of Cases (Percent)
No Specific Molecular Profile	218 (36.1%)
Mismatch Repair Deficient	287 (47.5%)
Polymerase-ϵ Ultramutated	30 (5.0%)
p53 Abnormal	69 (11.4%)
Age (years) (median (interquartile range))	68 (60–76)
Body mass index (kg/m^2^) (median (interquartile range))	27.3 (23.7–32.5)
Pelvic lymphadenectomy (number of cases, percent)	344 (57.0%)
Pelvic–aortic lymphadenectomy (number of cases, percent)	95 (15.7%)
Histology	
Endometrioid carcinoma	535 (88.6%)
Clear cell carcinoma	25 (4.1%)
Serous carcinoma	18 (3.0%)
Carcinosarcoma	13 (2.2%)
Undifferentiated carcinoma	13 (2.2%)
Grade (number of cases, percent) (For endometrioid only; n = 535)	
1	293 (54.8%)
2	155 (29.0%)
3	87 (16.3%)
Stage (number of cases, percent)	
IA	309 (51.2%)
IB	131 (21.7%)
II	47 (7.8%)
IIIA	33 (5.5%)
IIIB	6 (1.0%)
IIIC1	40 (6.6%)
IIIC2	18 (3.0%)
IVA	0 (0%)
IVB	20 (3.3%)
Adjuvant therapy (number of cases, percent)	
None	81 (13.4%)
Vaginal brachytherapy	281 (46.5%)
Whole pelvic radiotherapy	92 (15.2%)
Chemotherapy	24 (4.0%)
Chemotherapy + vaginal brachytherapy	41 (6.8%)
Chemotherapy + whole pelvic radiotherapy	85 (14.1%)

**Table 2 cancers-14-00651-t002:** Univariable Cox regression disease-specific survival analyses for ESGO/ESTRO/ESP 2021 clinicopathologic risk groups of endometrial carcinoma.

Molecular Subgroup	Low-Risk			Intermediate-Risk			High-Intermediate-Risk			High-Risk			Advanced/		
													Metastatic		
	N	HR	*p*	N	HR	*p*	N	HR	*p*	N	HR	*p*	N	HR	*p*
		(95% CI)			(95% CI)			(95% CI)			(95% CI)			(95% CI)	
Molecular subgroup			0.03			0.988			0.092			0.608			0.939
NSMP	114	1		33	1		33	1		32	1		6	1	
MMRd	119	1.7 (0.50–5.8)	0.397	48	1.3 (0.30–5.2)	0.758	52	3.5 (1.2–10)	0.024	61	1.3 (0.60–2.7)	0.536	7	1.1 (0.32–4.1)	0.841
*POLE*mut	21	Not calculable	0.982	3	Not calculable	0.987	5	Not calculable	0.984	1	Not calculable	0.977	0	-	-
p53abn	10	9.1 (2.0–41)	0.004	9	1.4 (0.15–14)	0.77	10	0.84 (0.094–7.5)	0.879	29	1.7 (0.77–4.0)	0.185	11	1.2 (0.39–4.0)	0.724

Abbreviations: CI, confidence interval; ESGO, European Society of Gynaecological Oncology; ESP, European Society of Pathology; ESTRO, European Society for Radiotherapy and Oncology; HR, hazard ratio; MMRd, mismatch repair deficient; NSMP, no specific molecular profile; *POLE*mut, polymerase-ϵ mutant; p53abn, p53 abnormal.

**Table 3 cancers-14-00651-t003:** Shift of patients between prognostic risk groups by molecular integrated classification schemas.

	Molecular Integrated Classification Schema				
Clinicopathologic Risk Group	ProMisE	N (%)	Leiden	N (%)	*p*
Low-risk (LR)	1 p53abn to IMR	10/264 (3.8%)	1 p53abn to IMR	10/218 (4.6%)	0.661
	9 p53abn to HR		9 p53abn to HR		
Intermediate-risk (IMR)	3 *POLE*mut to LR	10/93 (10.8%)	3 *POLE*mut to LR	10/77 (13.0%)	0.653
	7 p53abn to HR		7 p53abn to HR		
High-intermediate-risk	5 *POLE*mut to LR	15/100 (15.0%)	6 *POLE*mut to LR	16/88 (18.2%)	0.557
	10 p53abn to HR		10 p53abn to HR		
High-risk (HR)	1 *POLE*mut to LR	1/123 (0.8%)	2 *POLE*mut to LR	2/109 (1.8%)	0.603
Advanced/metastastic	-	0/24 (0%)	-	0/23 (0%)	0.602
All		36/604 (6.0%)		38/515 (7.4%)	0.341

Note: Pearson’s χ^2^ or Fisher’s exact test.

## Data Availability

All relevant data are within the paper.

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
