# Peer review of "Clinicopathologic vs. Molecular Integrated Prognostication of Endometrial Carcinoma by European Guidelines"

_cancers, 2022, doi:10.3390/cancers14030651_

Round 1

Reviewer 1 Report

Dear Authors,

thank you for submitting your manuscript to “Cancers” journal.

I think the manuscript offers some interesting updates on the relationship between clinicopathological and molecular classification of endometrial cancer.

Indeed, the integration of clinicopathological and molecular features is pivotal to define a more precise prognostic stratification of endometrial cancers, with promising therapeutical applications (I would recommend to see also PMID: 33754186, PMID: 32296930, PMID: 33715893, PMID: 31932106).

Furthermore, I have the following comments:

  • I think you should remove “The Cancer Genome Atlas” from keywords, it is not manuscript-specific
  • Authors cannot state that “POLE sequencing, the most laborous component of molecular classification, can safely be omitted in 60% of patients” basing on only one retrospective study, although it has a large sample size. The possibility to avoid POLE sequencing is an interesting issue, that should be investigated in further studies.

Reviewer 2 Report

The authors conducted a retrospective study on tumor tissue samples of 604 women with endometrial cancer, in order to verify whether the assignment of patients to the different risk groups proposed by the ESGO-ESTRO-ESP 2021 classification was modified by the knowledge of their molecular characterization in addition to the knowledge of their clinicopathological condition.

Thirty-six (6.0%) and 38 (7.4%) patients were shifted between risk groups by the ProMisE and Leiden molecular integrated schemas.

The authors conclude that POLE sequencing could be omitted in 60% of patients without affecting the risk group assessment.

The quality of the presentation is high as well as its scientific soundness, but the authors should better discuss with the readers the originality and the clinical significance of their findings.

Reviewer 3 Report

The authors reclassified a large, unselected retrospective collective of 604 endometrial cancer patients based on (i) clinicopathological and (ii) molecular risk groups according to the recent published ESGO-ESTRO-ESP guidelines. Noteworthy, recapitulation of molecular subgroups was performed by two distinct, validated classifiers, namely “ProMisE” and the “Leiden classifier”. Interestingly, the authors evidenced that molecular re-classification was associated with a 6% shift between risk groups. In addition, the authors showed that POLE sequencing, which is often not possible to be performed, can be safely omitted in 60% of patients.

The study is well done and the paper is written in a concise manner. The method section described well the used methods and results show relevant data. The discussion and conclusion section is written comprehensively.

Some minor comments could be addressed by the authors:

  • Table 1 (Clinicopathologic data) shows information of performed adjuvant treatment. Chemotherapy + WPR was performed in 14.1% of patients. Were those patients treated by sequence or concomitant radiochemotherapy?
  • Over the last years abnormal L1CAM (CD171) was shown to be relevant in EC risk classification. Although not integrated in the recent ESGO-ESTRO-ESP guidelines, L1CAM staining could be helpful in decision-making, especially in the NSMP subgroups (e.g. Kommoss FKF et al. BJC 2018). Do the authors have data on L1CAM expression in their collective? If not, their opinion on the relevance of L1CAM in endometrial cancer risk stratification would be interesting and could be integrated in the discussion section.
  • The authors stated that within their collective MMR deficiency (MMRd) was associated with poor survival in clinicopathologic classified high-intermediate risk ECs. As one possible explanation the authors pointed out a poor response of MMRd endometrial cancers to adjuvant radiotherapy and cited a recent publication of the first author of the actual publication (Loukovaara M et al. Cancer Med 2021). This finding is in contradiction to data published by Reijnen C et al. Gynecol Oncol (2019) showing an improved disease-specific survival of MMR-deficient endometrial cancers. Could the authors comment on that?
